# Maximizing the reusability of gene expression data by predicting missing metadata

**Pei-Yau Lung**[1], **Dongrui Zhong**[1], **Xiaodong Pang**[2], **Yan Li**[3], **Jinfeng Zhang**[1]*

**1** Department of Statistics, Florida State University, Tallahassee, United States of America, **2** Insilicom LLC, Tallahassee, United States of America, **3** Department of Breast Surgery, Peking Union Medical College Hospital, Peking Union Medical College, Chinese Academy of Medical Sciences, Beijing, China

* jinfeng@stat.fsu.edu

**Data Availability Statement:** All the data used in this study, including those from the Cancer Genome Atlas (TCGA, https://portal.gdc.cancer.gov/) and Gene Expression Omnibus (GEO, https://www.ncbi.nlm.nih.gov/geo/, GSE20194,

## Abstract

Reusability is part of the FAIR data principle, which aims to make data Findable, Accessible, Interoperable, and Reusable. One of the current efforts to increase the reusability of public genomics data has been to focus on the inclusion of quality metadata associated with the data. When necessary metadata are missing, most researchers will consider the data useless. In this study, we developed a framework to predict the missing metadata of gene expression datasets to maximize their reusability. We found that when using predicted data to conduct other analyses, it is not optimal to use all the predicted data. Instead, one should only use the subset of data, which can be predicted accurately. We proposed a new metric called Proportion of Cases Accurately Predicted (PCAP), which is optimized in our specifically-designed machine learning pipeline. The new approach performed better than pipelines using commonly used metrics such as F1-score in terms of maximizing the reusability of data with missing values. We also found that different variables might need to be predicted using different machine learning methods and/or different data processing protocols. Using differential gene expression analysis as an example, we showed that when missing variables are accurately predicted, the corresponding gene expression data can be reliably used in downstream analyses.

## Author summary

Large volumes of gene expression data are available at public databases such as Gene Expression Omnibus (GEO) and sequence read archive (SRA). They can be reanalyzed to solve previously infeasible biological problems. However, reanalysis studies using public genomics data have been hindered by the lack of necessary metadata for the analyses. This can be addressed by predicting the metadata using the gene expression data, which can then be used in the desired reanalysis with predicted metadata. This represents a new approach to increase the reusability of public gene expression data. Our study attempts to systematically investigate how this approach should be carried out. We found that one should not use all the gene expression data with metadata predicted for downstream analyses. While using all the gene expression data maximizes the sample size, the poorly predicted expression profiles may affect the quality of the downstream analysis. One needs to

GSE20271, GSE22093, GSE23988, GSE25055, GSE25065, GSE42822), are publicly available. The source code used in the study is available at https://github.com/dz16e/Reusability.

**Funding:** JZ is supported partially by a grant from National Institute of General Medical Science of National Institutes of Health, grant # R01GM126558. The funder had no role in the study design, data collection and analysis, decision to publish, or preparation of the manuscript.

**Competing interests:** Jinfeng Zhang is the founder and CEO of Insilicom LLC. Xiaodong Pang is the CTO of Insilicom LLC. Other authors have declared that no competing interests exist.

strike a balance between the amount of data included in the downstream analysis and the accuracy of predicted metadata. To address this problem, we designed a new metric called Proportion of Cases Accurately Predicted (PCAP), which is optimized in our specifically-designed machine learning pipeline. Using differential gene expression analysis as an example, we showed that when missing variables are accurately predicted, the corresponding gene expression data can be reliably used in downstream analyses.

This is a *PLOS Computational Biology* Methods paper.

## Introduction

Currently, large volumes of high-throughput genomic data are being generated in biomedical research every day by laboratories in both academia and industry. For example, as of May 23, 2018, the gene expression omnibus (GEO) database [1] consists of a total of 2,498,466 samples in 98,354 series generated by 18,519 different experimental platforms. Many federally-funded projects and initiatives are also generating unprecedented large volumes of genomic data, such as The Cancer Genome Atlas (TCGA, https://portal.gdc.cancer.gov/), the Genotype-Tissue Expression (GTEx) project [2], and the ENCyclopedia Of DNA Elements (ENCODE) project [3]. The availability of so-called biomedical Big Data has allowed new scientific discoveries to be made by mining and analyzing such data [4–10]. As high-throughput datasets are rich in information on cellular events, the same dataset can be reanalyzed alone or together with other data to address important questions that were previously not studied or not feasible due to limited availability of data [6, 7, 11].

However, the majority of public genomic data do not contain enough metadata, which severely limits their reusability[12]. Overcoming this limited reusability forms part of the FAIR (Findable, Accessible, Interoperable and Reusable) data principle [13]. For example, to understand the heterogeneity of breast cancer and to develop personalized treatment for breast cancer [11, 14], the biomarker information that determines the subtypes of breast cancer samples, such as estrogen receptor (ER), progesterone receptor (PR), and human epidermal growth factor receptor 2 (HER2) status, are normally required. Information on race is also necessary to study the racial disparity of breast cancer [6–8]. We did an analysis of available breast cancer gene expression data generated by platform GPL570 (Affymetrix Human Genome U133 Plus 2.0 Array) in the GEO database and found that there are 29,631 samples, of which ~85% do not have ER information, ~88% do not have HER2 information, and ~95% do not have race information. This high percentage of missing metadata severely limits what can be studied using these data. Accurate prediction of the missing metadata will substantially increase the reusability of the existing genomic data. However, as far as we know, no previous studies have shown that (1) gene expression data can be used to accurately predict a diverse set of missing metadata; and (2) the predicted metadata can be used reliably for downstream analyses.

There are two major strategies to predict missing metadata for existing genomic data. First, the traditional approach is to fill missing data by imputation, which uses the information of the same variable from other subjects in the same dataset to infer the missing data using relatively simple statistics [15]. For example, to fill the missing value for variable $i$ for subject $j$, one can use the mean value of variable $i$ from all the other subjects (or subjects similar to subjects $j$

based on other variables). Second, modern approaches based on machine learning technology are now widely used for inferring missing data in biomedical sciences with better performance than the traditional approach [15–19].

It has been shown that gene expression is very informative and highly predictive of various clinical outcomes, such as the status of biomarkers [20–26], tumor types/status [27–30], the risks of recurrence [31, 32] and survival [32–35], therapeutic response [11, 36–38], or other metadata [39–43]. For gene expression data, the gene expression profiles themselves, therefore, are ideally suited for inferring the missing metadata. It may seem quite straightforward to infer missing metadata using gene expression profiles, given the abundance of previous research works. However, if our goal is to recover the missing metadata and use such information to perform additional analysis, several issues need to be considered. First, we will likely want to exclude the data for which our prediction may not be accurate enough since the error will be transferred to downstream analysis. One may prefer the accuracy of the prediction to be at least above a certain threshold; Second, as genomic data are being generated by many different platforms, transferring models generated from one platform to other platforms is of concern [44]; Third, the existing metrics for evaluating machine learning methods, such as area under the curve (AUC), F1-score, precision, recall, and accuracy, are not ideal for such tasks as they aim to optimize the accuracy for whole datasets. A better objective would be to recover as much data as possible with accuracy above the threshold one prefers. Here accuracy can be defined by any proper metrics, such as AUC, F1-score, etc.. We believe this objective is better suited to maximize the reusability of public gene expression data when predicting missing metadata.

In this study, we investigated the above issues to infer missing metadata using multiple gene expression datasets with a wide variety of machine learning methods. We evaluated their performance on a total of 43 clinical variables to assess the accuracy level the current machine learning methods can achieve. We also investigated a robust normalization approach, rank normalization, on prediction performance. Models built using data generated by rank normalization are likely more transferable than other commonly used normalization methods, such as global normalization and quantile normalization [45, 46]. We introduced a new performance measure to evaluate the effectiveness of methods in recovering missing metadata, called Proportion of Cases Accurately Predicted (PCAP). PCAP is the percentage of data that can be predicted given a desired level of accuracy, where the actual accuracy measure (i.e. overall accuracy, precision, F1-score, etc.) can be defined by the researcher. $PCAP_{90}$ and $PCAP_{95}$ stand for the proportion of data that can be predicted with an accuracy of 90% and 95%, respectively. Through this study, we proposed a framework to select the optimal pipeline, which includes several components such as data processing, oversampling method, variable selection, machine learning model and choice of performance measures, for recovering missing metadata by maximizing $PCAP_{90}$ or $PCAP_{95}$. Using differential gene expression analysis (DGEA), we showed that gene expression data with predicted metadata can be reliably used for identifying differentially expressed genes between different groups defined by the predicted metadata. To achieve optimal performance, one should use the subset of data, which can be predicted with high accuracy, instead of using all the predicted data.

## Materials and methods

### Data

Gene expression data from both sequencing and microarray platforms were used in this study. The RNA-seq data from TCGA and GTEx were obtained from *recount*2 database [47, 48]. Each sample contains 20,483 RNA gene expression values and metadata, such as race, receptor

status, and tumor type. We used data from five cancer types: breast invasive carcinoma (BRCA), lung adenocarcinoma (LUAD), lung squamous cell carcinoma (LUSC), ovarian serous cystadenocarcinoma (OV), and prostate adenocarcinoma (PRAD) with totally 2954 patient samples to perform study on four example variables: race, ER status, HER2 status and PR status. An additional study was performed using all the TCGA data with 11284 samples on 42 variables to further validate the results obtained using the four representative variables. The microarray data were collected from 6 data series in the GEO database [1] generated by the GPL570 platform, which contains 879 breast cancer patient samples with 22,283 gene expression values in each sample.

For each variable, the subset of samples with missing values was not considered in the analysis of the corresponding variable, since we cannot evaluate the performance of our method if we use these samples. For race, we included only African American (AA) and Caucasian American (CA) samples to simplify the analysis and discussions. For ER, PR, and HER2 variables, only samples with positive or negative values were included, while samples with "unequivocal", "unknown", etc. values were excluded. In TCGA data, ER, PR and HER2 status were only available for breast cancer patients ($n$ = 1246). In the microarray data, in addition to ER, PR, and HER2 variables, we chose treatment response instead of race because of the unavailability of race data. There are two possible values for treatment response: pathological complete response (pCR) and residual disease (RD). Fig 1 shows the percentages of each possible value for the clinical variables used in this study. We assigned binary values to each clinical variable where we labeled the value with less number of cases as +1 and the value with more number of cases as -1 (or 0 depending on the convention used in a machine learning algorithm). For example, for race, we labelled AA as +1 and CA as -1.

## Normalization for gene expression

Three normalization methods were used in this study: read per million (RPM) normalization for next generation sequencing (NGS) data, quantile normalization for microarray data and rank normalization (RN) for both NGS and microarray data. For RN, each gene expression value was replaced by its rank, ranging from 1 (lowest) to $n$ (highest), within a patient. If there

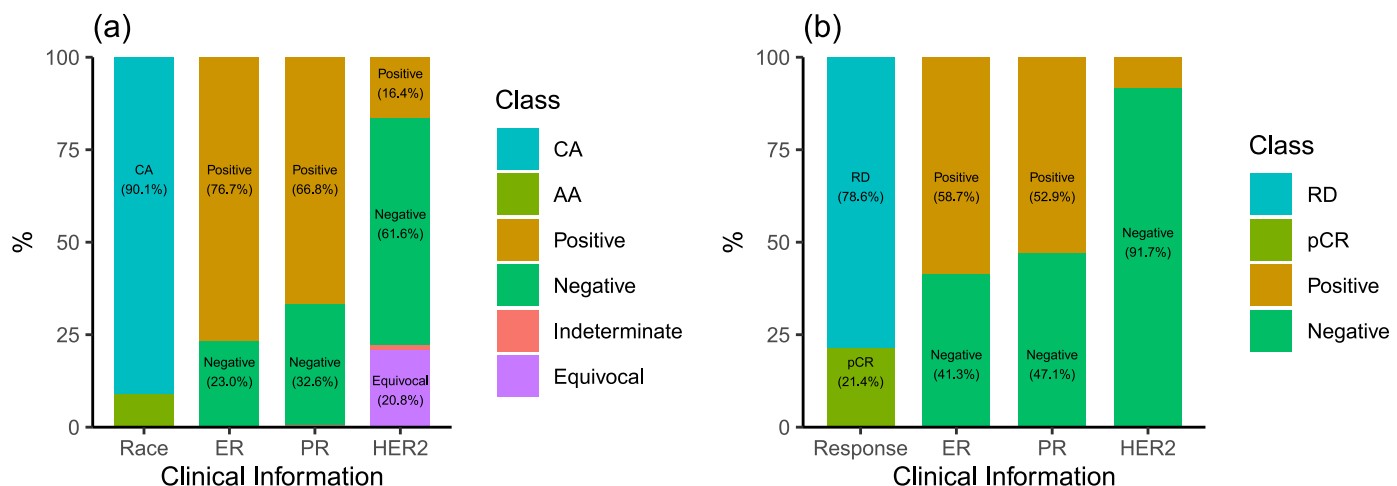

**Fig 1.** The number of patients in each class of four clinical variables for both sequencing data from TCGA (a) and microarray data from GEO (b). Numbers in parenthesis are the percentages of the corresponding class. AA: African American. CA: Caucasian American. pCR: pathological complete response. RD: residue disease.

was a tie for two or more values, the average of these values was used as the rank. The ranks were rescaled by dividing by the sum of total ranks and multiplying by $10^6$.

## Oversampling methods

There are many oversampling methods based on different assumptions and goals [49–51]. We chose the Synthetic Minority Oversampling Technique (SMOTE) [49] in our study. Rather than oversampling with replacement, SMOTE increases the number of the minority class by creating synthetic samples. These synthetic samples are generated from the information of minority samples. With the minority and synthetic samples, classifiers will build larger and more general decision regions, which tend to give more accurate predictions [49–51]. SMOTE can be applied from the Imbalanced-learn package in Python [52].

## Models

In this section, we consider a binary classification setting. We define a model as a combination of a machine learning algorithm, number of genes selected, oversampling choice, and data normalization choice. The machine learning algorithms used in this study include XGBoost [53], random forest (RF) [54], support vector machines (SVM) [55], and LASSO [56]. Tuning parameters were optimized from 10 sets of random combinations using 3-fold cross validation. To select the most predictive genes, a Welch two-sample t-test was first conducted to find differentially-expressed genes between two classes at the significance level of 0.1, where the two classes correspond to the two binary values of the variables to be predicted. Next we use recursive feature elimination (RFE) [57] to select an optimal set of genes (smaller than 100) or a predefined number (10 or 25) of genes using 3-fold cross validation. We restricted the optimal number of genes to less than or equal to 100.

## Evaluation measures and optimization procedure

To evaluate the prediction performance of each model, three measures were used: Area under the receiver operating characteristic curve (AUROC), $F_1$-score, and our proposed new measure PCAP. In this study, $PCAP_x$ stands for the percentage of cases which can be predicted with x% precision in a model. Other performance measures other than precision can also be used in practice, such as accuracy, recall, or F1-score. The $PCAP_x$ is calculated from a 10-fold cross validation each with 90% of the data as training and 10% as testing. For each of the 10 folds, a $k$-fold ($k = 10$ in this study) cross validation was further performed resulting in a smaller training dataset and a validation dataset in each fold. The predicted probabilities of the validation set were first sorted. Each percentile of the predicted probability (from 50 to 99) was used as a cutoff to calculate the precision and recall in the validation set. If a percentile results in a precision greater than or equal to x%, than it would be labeled as 1 (otherwise 0). The smallest percentile with most 1s (up to $k$) among the $k$ folds was selected as the cutoff and the corresponding average recall was recorded. After the 10-fold cross-validation, the average of the selected percentile from each $k$-fold cross-validation was chosen as the cutoff value for the test dataset, and the average recall is used as the estimated $PCAP_x$ for the test dataset.

The 10-fold cross validation in the training data helps to find the cutoff that results in the desired precision. The returned value of recall is the $PCAP_x$, which is an estimate of the recall value in the unseen test data given x% of precision. In this study, we use $PCAP_{90}$ for the assessment of different models.

Another benefit of performing cross validation in the training data is for model selection. Given various models, we select the model that has the best median PCAP, instead of the

maximum PCAP, to achieve a more robust estimate of the average PCAP value. This selected model is most likely to generalize well when recovering missing metadata in future datasets.

## Results

### Predictive performance in sequencing data

The performance of models was tested by stratified 10-fold cross validation. Table 1 shows the median of AUROC values across 10 folds under each model. The AUROC value reached up to 0.987 when predicting race, and reached over 0.9 when predicting ER and PR status. When predicting HER2 status, the AUROC value reached 0.797. The $F_1$-scores (Table A in S1 Material) confirm that these clinical variables can be predicted with good accuracy. Models with small, pre-defined number of genes can obtain good performance as well. Tables B and C in S1 Material show that using 10 or 25 genes results in satisfactory results. There is no single model that performs the best for all four variables.

### Predictive performance in microarray data

The medians of AUROC values across 10 folds under each model (Table 2) indicate the models can also predict clinical variables well using gene expression data generated by microarray experiments. The AUROC values obtained are as high as 0.918 for pCR, 0.970 for ER, 0.947 for PR, and 0.938 for HER2. Table D in S1 Material shows the F1-scores of different models and settings. For predicting ER, PR, and HER2 status, it is possible to get comparative performance using fewer number of genes, which are shown in Tables E and F in S1 Material. Again, the results show there is no single model that performs the best for all four variables.

### Effect of rank normalization

Gene expression data are usually generated by many different experimental platforms. Currently, at the gene expression omnibus (GEO) database [58], there are 15 major platforms measuring gene expression (mRNA values), each with more than 10,000 samples, together with hundreds of minor platforms with smaller number of samples. Using commonly used normalization methods, such as quantile and global normalization, models developed using

**Table 1. AUROC values from different algorithms for variables in sequencing data.** RN denotes that the predictors were rank-normalized. SMOTE denotes that the samples were balanced by Synthetic Minority OverSampling Technique. An optimal number of genes were selected by recursive feature elimination with cross validation.

| | Race | | | ER[a] | | |
|---|---|---|---|---|---|---|
| | **RPM** | **RN** | **RN + SMOTE** | **RPM** | **RN** | **RN + SMOTE** |
| LASSO | 0.946 | 0.979 | 0.982 | 0.895 | 0.915 | 0.906 |
| Random Forest | 0.946 | 0.931 | 0.975 | 0.949 | 0.949 | **0.957** |
| XGBoost | 0.982 | 0.975 | 0.985 | 0.947 | 0.947 | 0.948 |
| SVM | 0.835 | **0.987** | 0.975 | 0.834 | 0.908 | 0.904 |
| | PR[b] | | | HER2[c] | | |
| LASSO | 0.836 | 0.883 | 0.875 | 0.701 | 0.791 | 0.785 |
| Random Forest | 0.909 | **0.920** | 0.902 | 0.788 | 0.793 | **0.797** |
| XGBoost | 0.899 | 0.900 | 0.899 | 0.793 | 0.784 | 0.788 |
| SVM | 0.789 | 0.858 | 0.863 | 0.702 | 0.792 | 0.790 |

a: Only ER-positive (833) or ER-negative (243) patients were included.

b: Only PR-positive (726) or PR-negative (347) patients were included.

c: Only HER2-positive (173) or HER2-negative (581) patients were included.

**Table 2.  AUROC values from different algorithms for variables in microarray data.**

| | Response | | | ER[a] | | |
|---|---|---|---|---|---|---|
| | **RPM** | **RN** | **RN + SMOTE** | **RPM** | **RN** | **RN + SMOTE** |
| LASSO | 0.910 | 0.899 | 0.875 | 0.954 | 0.960 | 0.959 |
| Random Forest | **0.918** | 0.909 | 0.914 | 0.968 | **0.970** | 0.968 |
| XGBoost | 0.884 | 0.883 | 0.899 | 0.968 | 0.967 | 0.960 |
| SVM | 0.904 | 0.889 | 0.892 | 0.952 | 0.955 | 0.960 |
| | PR[b] | | | HER2[c] | | |
| LASSO | 0.938 | 0.939 | 0.941 | **0.938** | 0.924 | 0.901 |
| Random Forest | 0.937 | 0.936 | 0.942 | 0.916 | 0.906 | 0.894 |
| XGBoost | 0.938 | 0.938 | 0.933 | 0.894 | 0.880 | 0.917 |
| SVM | **0.947** | 0.928 | 0.927 | 0.931 | 0.909 | 0.896 |

a: Only ER-positive (833) or ER-negative (243) patients were included.

b: Only PR-positive (726) or PR-negative (347) patients were included.

c: Only HER2-positive (173) or HER2-negative (581) patients were included.

data from one platform may not be generalizable to data produced by other platforms [59]. Developing different models for data generated by different platforms may not be an ideal solution for this problem. Rank normalization (RN) is a robust normalization method since only normalized rank information is kept in the normalization process. Most normalization methods do not change the rank (relative order) of the gene expression values within a sample. Here we investigate whether RN can be used as a robust normalization method for building machine learning models, which will yield better transferability of the resulting models.

To that end, we subtracted the $F_1$-score of either RPM (sequencing data) or Quantile normalization (microarray data) from the $F_1$-score of RN of each fold and took the average of the differences from each fold. As can be seen from Fig 2, RN helps to improve the performance of LASSO and SVM in sequencing data while giving comparable results in other cases. Predictions using rank-normalized gene expression values got higher $F_1$-scores than using RPM normalization (Fig 2 and Table A in S1 Material). It is interesting to note that RN has comparable or better performance than commonly used normalization methods (RPM for sequencing data and Quantile normalization for microarray data) although it loses some valuable information during the normalization process.

## PCAP$_{90}$ after model selection

Table 3 shows the predictive performance in terms of PCAP$_{90}$. Numbers are the average PCAP$_{90}$ of 10-fold cross validation, where in each fold the model was selected by our model selection pipeline. All models were trained by maximizing either the $F_1$-score or PCAP$_{90}$, respectively. As shown in Table 3, models trained by optimizing PCAP$_{90}$ obtained higher PCAP$_{90}$ than optimizing $F_1$-score in both sequencing and microarray data.

## Applying the framework to a large number of variables

We then applied our framework to a much larger set of variables to illustrate the generality of the pipeline we designed and the conclusions made using five example variables. We downloaded the TCGA data from *recount2* project [47], with totally 11284 samples where each sample has over 800 metadata variables. We selected 42 categorical variables, which are likely predictable using gene expression data. For completeness, these 42 variables included four

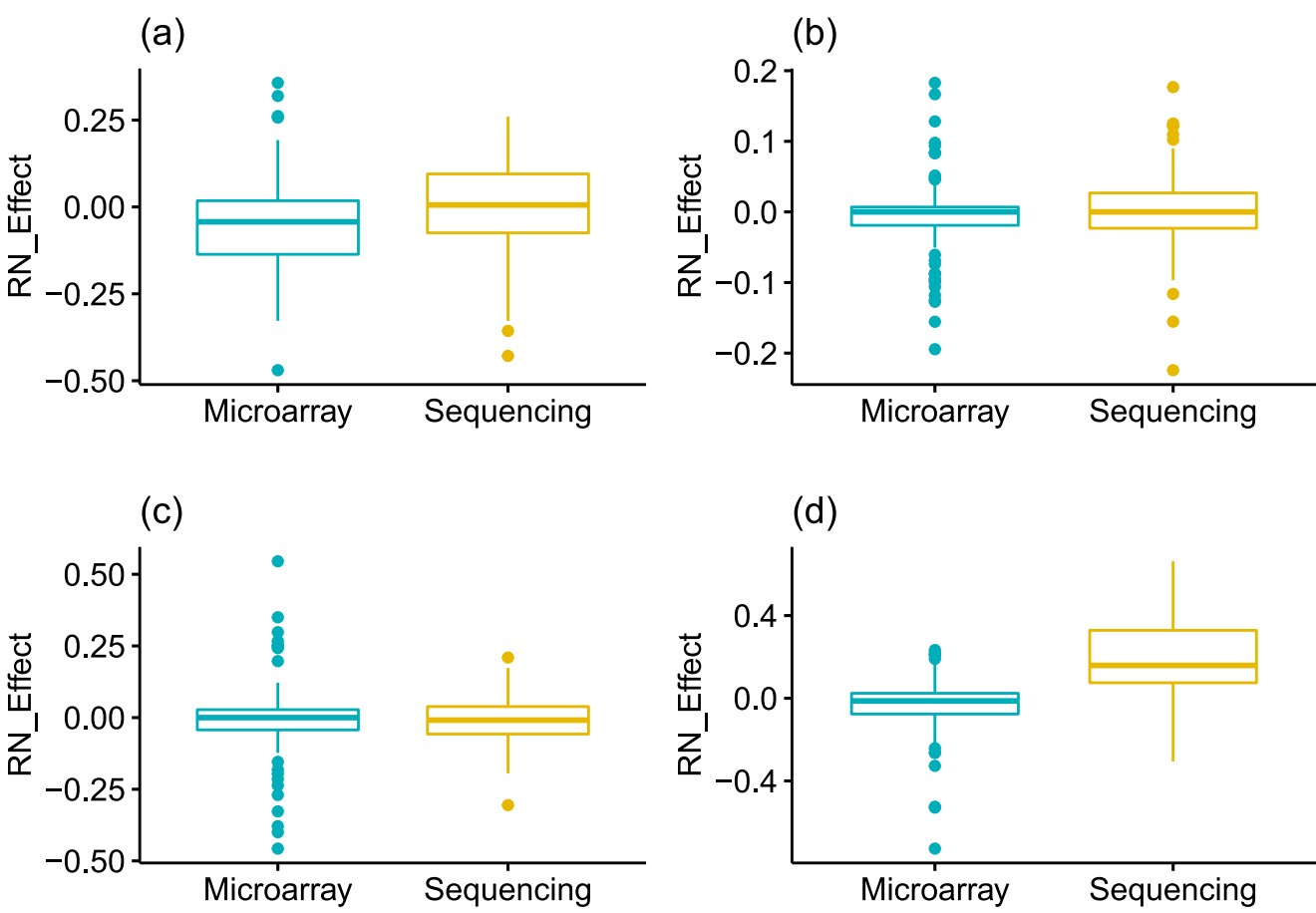

**Fig 2. Comparison of rank normalization (RN).** RN_Effect denotes the $F_1$-score from RN minus $F_1$-score from RPM or quantile normalization of each fold using: (a) LASSO, (b) Random Forest, (c) XGBoost, and (d) SVM. For LASSO and SVM, RN gave better performance for sequencing data. For other models, RN gave comparable performances.

variables used early on (race, ER, HER2 and PR status). Continuous variables were not included in this study. They can be included as categorical variables by discretization.

The number of available samples (ones with non-missing values) varies greatly for different variables. For each variable, we only selected top two classes with the most samples, so that we

**Table 3. PCAP$_{90}$ and Precision after model selection (mean).** The numbers are the average of 10-fold cross-validation. By optimizing PCAP$_{90}$, we can generally achieve better PCAP$_{90}$ values (and/or better precision) compared to optimizing F1-scores. In all cases, by optimizing PCAP$_{90}$, we can achieve the desired precision of 90%. However, by optimizing F1-scores, this cannot be always achieved (colored as red).

**Sequencing data**

|  | Race | ER | PR | HER2 | Average |
|---|---|---|---|---|---|
| Optimizing $F_1$-score | 0.217 (0.863) | 0.167 (0.858) | 0.385 (0.903) | 0.248 (0.983) | 0.254 (.902) |
| Optimizing PCAP$_{90}$ | 0.216 (0.98) | 0.210 (0.90) | 0.360 (0.962) | 0.260 (0.915) | 0.262 (.939) |

**Microarray data**

|  | Response | ER | PR | HER2 | Average |
|---|---|---|---|---|---|
| Optimizing $F_1$-score | 0.056 (0.85) | 0.686 (0.926) | 0.570 (0.953) | 0.220 (0.95) | 0.383 (0.920) |
| Optimizing PCAP$_{90}$ | 0.09 (0.95) | 0.695 (0.954) | 0.553 (0.966) | 0.250 (0.95) | 0.397 (0.955) |

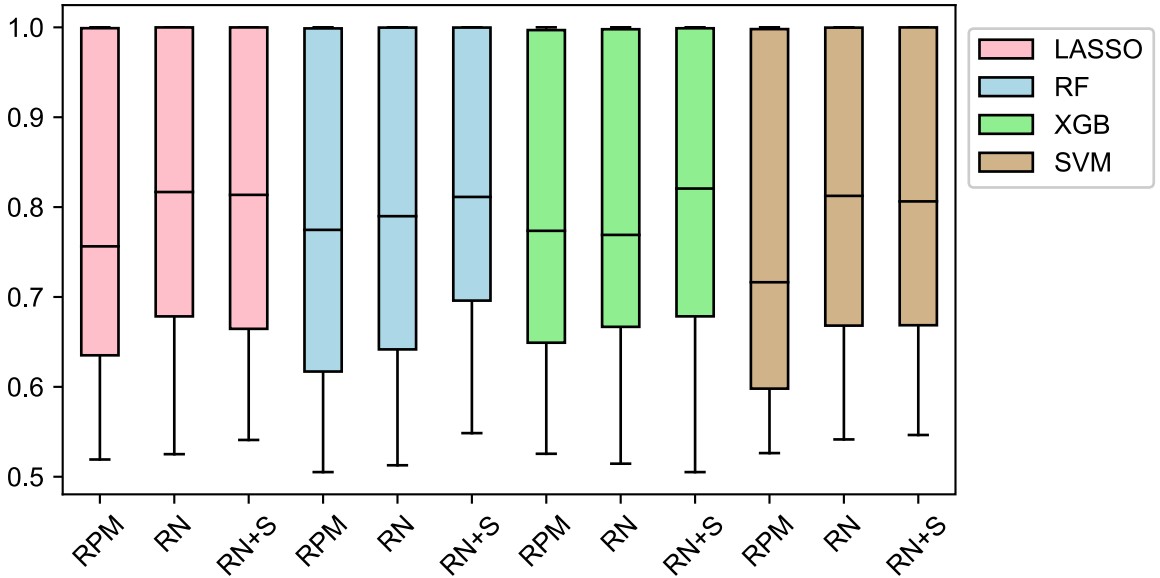

**Fig 3. AUROC of different combination of models and normalization methods.** RN+S means we apply both rank normalization and SMOTE. Each box contains 42 AUROC values for 42 variables. Each AUROC value is the mean of the 10-fold cross validation. In general, applying RN and SMOTE shows improvement over RPM.

have a binary classification setting. Our pipeline can be extended to multi-class predictions in the future (i.e. formulating a multi-class prediction problem as multiple one-vs-rest binary prediction problems). We applied the same framework described earlier and recorded the detailed model performances in the Supplementary file. Fig 3 shows the performance of different models and the effect of rank normalization (RN). We can see a very large variation in the predictability of variables using gene expression data, which is reflected by the variation of AUROC values among different variables. Variables such as "tissue site" and "clinical stage" could be predicted with very high AUROC values, while variables like "alcohol history" and "white cell count" gave rather low AUROC values (around 0.6).

Another interesting issue we would like to investigate is the transferability of models between different datasets[40]. That is, whether we can achieve similar performance using models trained on one dataset to predict metadata of another independent dataset. To this end, we used samples from GTEx project downloaded from recount2 database. The first experiment is using models trained on TCGA lung tissue samples to predict race and gender on TCGA brain tissue samples. For race, there are 854 white and 92 black or African American in the lung tissue samples, and 641 white and 33 black or African American in the brain tissue samples. For gender, there are 468 female and 680 male samples in the lung tissue samples, and 298 female and 402 male samples in the brain tissue samples. We performed binary classifications using the same framework as described early on. Table G in S1 Material compared the performance of the models.

The second experiment is using models trained on TCGA data (1246 breast tissue samples and 1156 lung tissue samples) to predict tissue site information on GTEx data (218 breast tissue samples and 374 lung tissue samples). We again applied the same framework and the results are shown in Table H in S1 Material. From Table G and Table H in S1 Material, we can see that the accuracy on race and tissue site are quite high when predicted by models built using different datasets. For dataset generated by the same platform and normalized using the

same procedure, we expect the models are quite transferable. However, for datasets generated using different platforms, one should use a cross-platform normalization method to increase the transferability of the models [44].

## Using predicted information in statistical inference

**Comparison between using true metadata and using predicted metadata.**   The goal of predicting missing metadata is to use the corresponding gene expression data with predicted metadata in other analyses. We investigated the usefulness of the predicted data in downstream analyses using differential gene expression analysis (DGEA), a very common task for gene expression data analysis. We performed DGEA between Caucasian (CA) and African American (AA) breast cancer patients using two-sample t-tests (after log2 transformation of the data) based on the race information provided in TCGA and the race information predicted using our method. Totally we have 2221 patient tissue samples who are either CA (1994) or AA (227), which was divided into ten folds with approximately equal number of samples in each fold. The training data (Dataset1) has 8 folds, test data (Dataset2) has 1 fold, and the last fold was treated as a newly collected dataset (Dataset3, more details later). We first performed DGEA in Dataset1 by randomly sampling 1 fold of data (with the same CA:AA ratio) to generate the list of differentially expressed genes (DEGs), called DEG1. We then conducted DGEA for Dataset2 using their true race information to obtain another list of DEGs, called DEG2_t. The third DEGA was performed using Dataset2 again by using predicted race, where the predictive model was built using Dataset1. This gave DEG2_p. The fourth DGEA was done using Dataset3 to produce DEG3. In this comparison, we assume that we can collect some new gene expression data with race information and perform the same DGEA. The last DGEA was done by first randomly permuting the race in Dataset2 to generated DEG2_r. We compute the number of overlapping DEGs between DEG1 and DEG2_t, between DEG1 and DEG2_p, between DEG1 and DEG3, and between DEG1 and DEG2_r. Computing the number of overlapping DEGs was done iteratively for each of the ten folds, then the average number of overlapping DEGs was computed. We repeated the process 50 times and drew the boxplots for the average number of overlapping DEGs in Fig 4.

Significance tests among the five groups (True, PCAP$_{95}$, PCAP$_{90}$, CollectNew, Random) in Fig 4 showed that when computing the number of overlapping DEGs, there is no significant difference between using true race information and using predicted race (p-values between True and PCAP$_{95}$ or PCAP$_{90}$ are 0.28 and 0.54, respectively). There is no significant difference between using predicted race and using the true race of newly collected data, either (p-value between PCAP$_{90}$ and CollectNew is 0.09). Randomly permuting the race will make the result significantly different (p-value between True and Random is smaller than $10^{-38}$). This indicates that datasets with metadata predicted using our method can be used reliably in the downstream analysis.

**Effect of including less accurate predictions to the quality of downstream analyses.**   So far, we have been assuming that we would like the prediction to be at least above certain accuracy. Is this necessary? Should one use all the predicted data instead? To answer these questions, we looked at the performance of DGEA when using subsets of data with different levels of accuracy. When using subset of data which can be predicted with high accuracy, the sample sizes are smaller compared to using all the predicted data. It is therefore not obvious that we should necessarily use the subset of samples which can be predicted accurately. Here we assumed that the actual accuracy of a prediction is positively correlated with the predicted probability, which is quite reasonable and has been observed in previous studies [11]. This means that in a binary classification problem, the model tends to have higher accuracy on samples with predicted probability close to 0 or 1 (as compared to samples with predicted

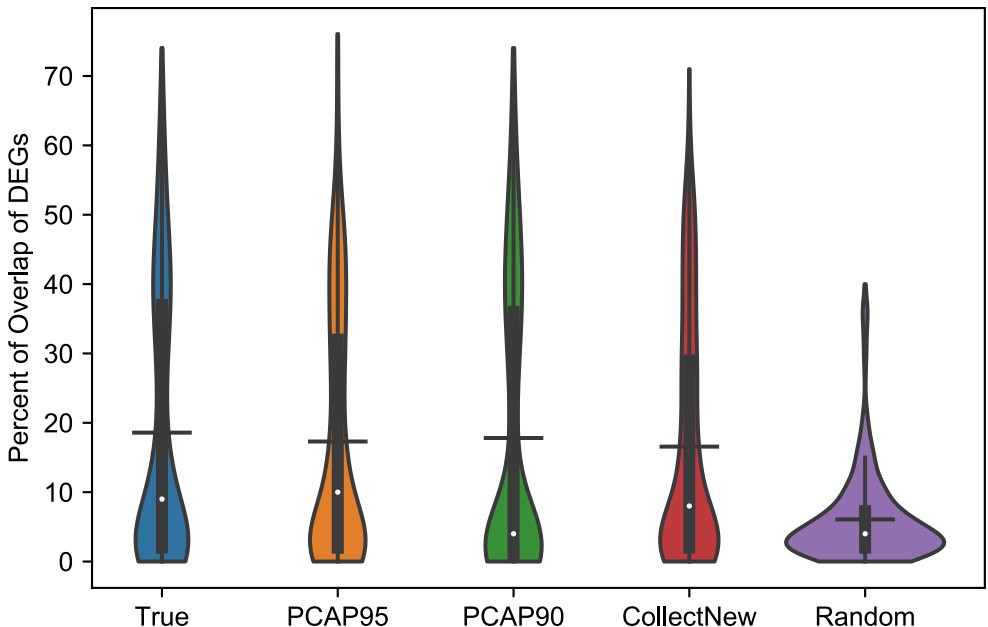

**Fig 4. Violin plot for the overlap of differential expressed genes (DEGs) detected by differential gene expression analyses using five different datasets.** The numbers are averages of ten runs. The ticks in the violin plots are means and the white dots are medians. True: DEGs obtained using the true race information in the original TCGA data; PCAP$_{95}$: DEGs obtained using a dataset where race is predicted by optimizing PCAP$_{95}$; PCAP$_{90}$: DEGs obtained using a dataset where race is predicted by optimizing PCAP$_{90}$; CollectNew: DEGs obtained by collecting a new dataset with known race information. This dataset is part of the TCGA data which was not used in other analyses; Random: DEGs by randomly permuting the race assignments among the patients in the dataset. Significance tests among the five groups: p-values between True and PCAP95 or PCAP90 are 0.28 and 0.54, respectively; p-value between True and CollectNew is 0.09; p-values between CollectNew and PCAP95 or PCAP90 are 0.51 and 0.30; p-value between True and Random is smaller than $10^{-38}$.

probability around 0.5). We can use predicted probabilities to select subsets of samples with different levels of accuracy.

We applied the same general framework as described in Method section to the following experiment. For a binary variable, its true values give a true group separation (Grp1, Grp2), and the predicted values give a predicted group separation (Grp_p1, Grp_p2). Consider the following two kinds of DEGA:

- First DEGA uses the true group separation (Grp1, Grp2). This serves as the ground truth. Here all samples are used. We call it DEGA_t.

- Second DEGA uses the predicted values. We chose a cutoff value $c$, and Grp_p1 contains only samples with predicted probability within $[0,c)$, and Grp_p2 contains only samples with predicted probability within $[1-c,1]$. Other samples will not be used in this DEGA. We call it DEGA_p($c$).

Using the above setting, we performed binary classifications on 42 variables using TCGA data from *recount2*. For each variable, we first performed DEGA_t using all samples, and got the true differentially expressed gene list DEG_t. Then we used the predicted probabilities from 10-fold cross validation. For each of the 10 fold, we performed DEGA_p($c$), where $c = 0.05, 0.1, 0.15,...,0.5$. Therefore, for each $c$, we obtained 10 lists of differentially expressed genes DEG_p($c,i$), $i = 1,2,...,10$. The numbers from the 10 lists were averaged. We then computed the percentage of overlapping DEGs between DEG_t and DEG_p($c,i$). In Fig 5, we used

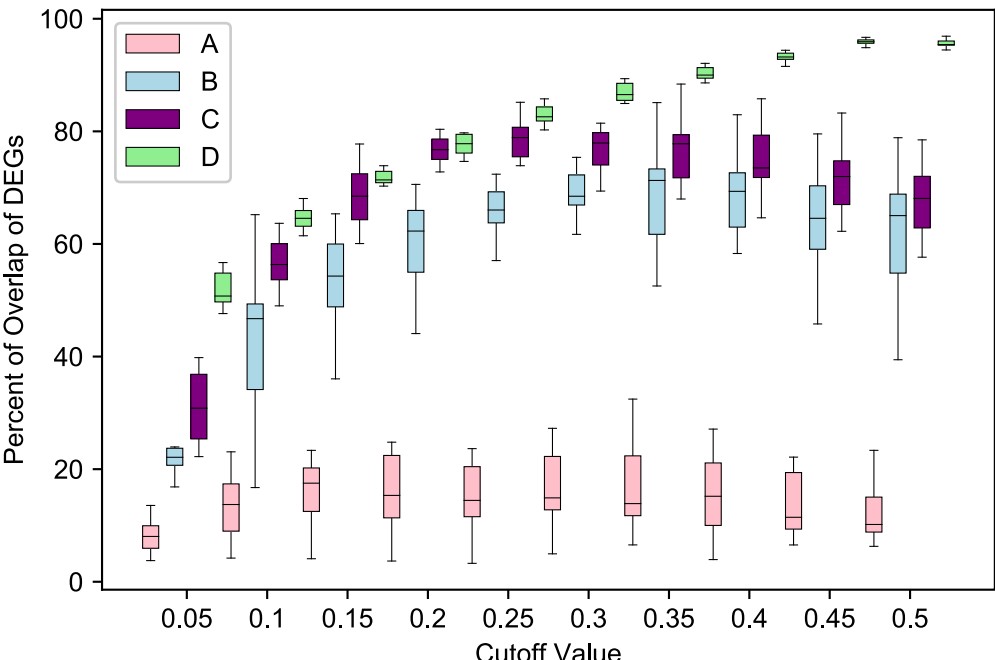

**Fig 5. Boxplot for the overlap of differential expressed genes (DEGs) between results using true metadata and predicted metadata using different accuracy cutoffs.** X-axis is the accuracy cutoff value. Cutoff value of 0.05 means we will use the samples whose predicted probabilities (for one of the class) are either less than 0.05 or greater than 1–0.05 (0.95). Samples with predicted probabilities less than 0.05 corresponds to one of the two classes, while samples with predicted probabilities greater than 0.95 correspond to the other class. The two classes are used to define the two groups in differential gene expression analysis. For variable B and D, we can see that the optimal performance (maximum percent of overlap of DEGs) were achieved around 0.35 and 0.25, respectively. This figure showed that both accuracy of the predicted data and sample sizes are important. Low cutoff values correspond to high accuracy, but smaller sample sizes, while high cutoff values correspond to low accuracy but larger sample sizes. The four variables are from TCGA dataset in *recount2*: A: cgc_case_performance_status_score_karnofsky. B: xml_tobacco_smoking_history. C: xml_breast_carcinoma_progesterone_receptor_status. D: gdc_cases.project.primary_site. Each box plots 10 values from the 10-fold cross validation.

four different variables from TCGA dataset in *recount2* (A: cgc_case_performance_status_s-core_karnofsky, B: xml_tobacco_smoking_history, C: xml_breast_carcinoma_progestero-ne_receptor_status and D: gdc_cases.project.primary_site) to show how different choices of $c$ affect the overlap of DEGs. Also, Fig 6 gives a summary of results for all 42 variables, where we plotted the best $c$ and the corresponding highest overlap percentage (median of the 10-fold cross validation) achieved. Smaller $c$ means samples in (Grp_p1, Grp_p2) will have predicted probabilities closer to 0 or 1, and thus higher accuracy. This will increase the accuracy of used samples, but reduce the sample size in DEGA. The performance of DEGA will be affected by two factors: the accuracy of predicted variable that defines the group separation, and the number of samples in each group. In Fig 5, the best cutoff value for variable D is .45 or .5, which means one can use all the predicted data for DGEA. For variable B and C the optimal cutoff values are 0.35 and 0.25, respectively, indicating that one should use only those samples whose variables can be accurately predicted to achieve the optimal performance in DGEA. For variable A, it may not be a good idea to use gene expression data to predict its values. Fig 6 shows the optimal cutoff values and the corresponding overlaps that can be achieved for all the 42 variables included in this study. It clearly showed that for most of the variables, one should not use all the predicted data, which validated the overall assumption of our study.

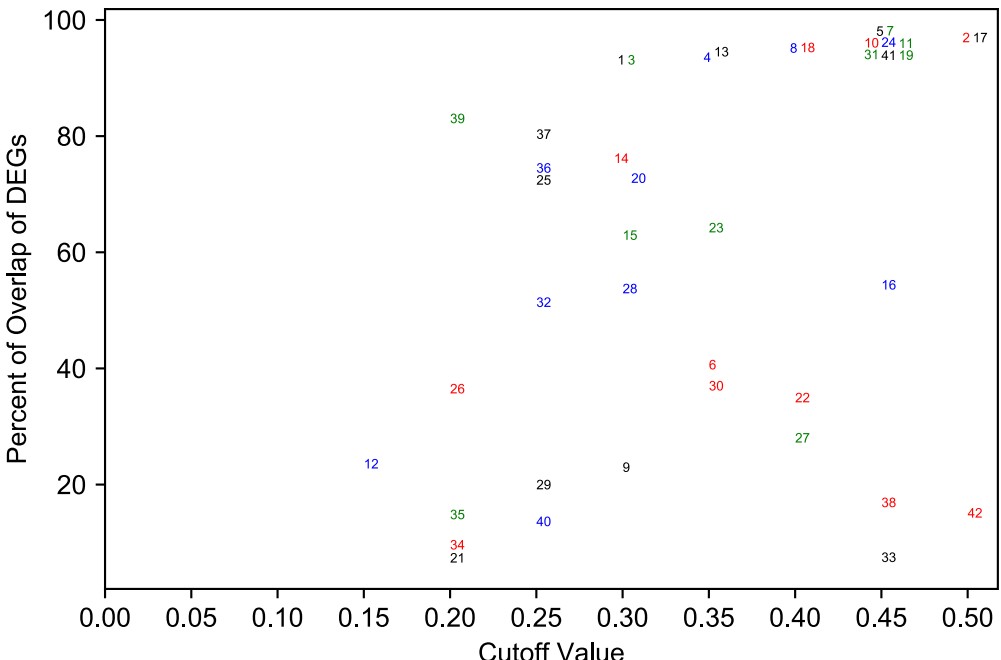

**Fig 6. The best cutoff value and the highest percentage of overlap of DEGs for 42 metadata variables in TCGA data.** Each point represents one variable (variable name can be found in the supplementary file Figure6_variable_names.xlsx). It shows the best cutoff value *c* and the highest median percentage of overlap from the 10-fold cross-validation. The cutoff value and percent of overlap of DEGs correspond to the optimal cutoff values and highest percent of overlap in Fig 5. For variable B in Fig 5, its value will be 0.35 and ~70 in this figure.

## Discussions

Genomics data are not useful if necessary metadata are not available. Unfortunately, a large amount of such important metadata is missing in public genomics datasets. In this study, we proposed a framework for maximizing the reusability of public gene expression data by predicting the missing metadata using machine learning methods. To develop and validate the framework, we used microarray and sequencing gene expression data with more than 10,000 cancer patient tissue samples to investigate whether we can predict missing metadata, such as race, ER, PR, HER2 and treatment response. Using a total of 43 variables (42 from TCGA and 4 from GEO), our study has shown that gene expression profiles can be used to predict metadata accurately. Out of over 20,000 genes, we can select small numbers of genes to obtain reliable predictions. For those variables for which reliable predictions cannot be achieved for all the missing metadata, we can select a subset of reliable predictions using our pipeline and a new measure designed for maximizing the reusability of public gene expression data. We found that different variables require different methods and parameter settings to achieve optimal performance. This is consistent with the well-known notion that no single method is the best for all kinds of machine learning tasks.

In addition to machine learning algorithms, normalization methods can have a substantial effect on the performance as well. In this study, we found that the robust rank normalization (RN) [60] can produce better or comparable performance than commonly used normalization methods (RPM for sequencing data and quantile normalization for microarray data). Predictions from rank-normalized sequencing data resulted in higher $F_1$-scores. In microarray data, RN gave comparable performances. RN is less sensitive to experimental noise and outliers. It

also allows researchers to conveniently combine several datasets generated from different platforms. Our result indicates that RN can be a good choice when building machine learning models either for predicting metadata to maximize the reusability of public gene expression data or for general model building purposes using gene expression data.

Successful statistical models depend on the quality of the data used for building the models. The accuracy of the predicted values for the missing data needs to be carefully evaluated to ensure the quality of the data to be used in downstream applications. While traditional evaluation metrics can be used, they are not ideal because they aim for accurate predictions of whole datasets. When researchers try to reuse public genomics data, they do not need to use all the data. Additionally, the accuracy for the whole dataset may not reach the desired accuracy threshold. They can use only the part of the data for which missing metadata can be reliably predicted. Here, a reasonable metric to optimize would be, give a certain accuracy threshold, the proportion of data that can be predicted above that threshold. The higher the number, the more data we will have for reuse with desired accuracy.

With the above reasoning, we proposed the metric Proportion of Cases Accurately Predicted (PCAP) for the purpose of maximizing reusability of public gene expression data. In addition, we proposed a selection pipeline to select a model from various combinations of algorithms, normalization methods, and data balancing procedures. Our results showed that we were able to recover a high percentage of samples with the desired accuracy. It is also recommended that one should maximize PCAP, instead of traditional performance measures for the whole dataset, when building models to obtain higher percentages of usable samples.

We also demonstrated the effectiveness of the predicted metadata in downstream inference tasks. In the study, we performed differential gene expression analysis (DGEA) using predicted race and found that the effectiveness of the analysis using predicted metadata is similar to that using true metadata. This demonstrated that our framework for maximizing the reusability of gene expression data can be reliably used in the future by other researchers. The larger study using 42 variables also indicated that we may not need to use a high accuracy cutoff to select subset of samples for downstream analyses. From Fig 6, $PCAP_{70}$ may be a value that strikes a good balance between accuracy and sample sizes.

There is a possibility that the subset of the data that can be predicted with high accuracy is systematically different in some way from the whole dataset. In such cases, the conclusion one can draw will be limited to only the subset of the data that can be predicted with high accuracy. In real clinical settings, for example biomarker discoveries, one can use the predictive model to stratify patient population and only apply the discoveries (i.e. identified diagnostic biomarkers) to that population.

It is worth noting that the metadata we have used as true information may have some noise in them. For example, it is well acknowledged that the self-reported race/ethnicity has high inaccuracy levels [61, 62]. These inherent errors will limit the upper bound of the accuracy we can achieve. From Fig 6, we can see that not all the metadata can be predicted accurately. When applying our method, the best practice would be doing training and validation on observations with no missing values, then predicting missing values using the trained models. If it appears that a variable can be predicted well based on the performance on validation data, then the framework can be used to predict the variable and use the predicted values for downstream analyses.

## Supporting information

**S1 Material.** Supplementary text that include all the supplementary tables (Tables A-H). (DOCX)

**S1 Data. The detailed machine learning results for all the variables and their descriptions.**
(XLSX)

**S2 Data. The mapping between points in Fig 6 and variable names.fsf.**
(XLSX)

## Acknowledgments

The authors thank the scientists and funding agencies for generating the data used in this study.

## Author Contributions

**Conceptualization:** Jinfeng Zhang.

**Data curation:** Pei-Yau Lung, Dongrui Zhong.

**Formal analysis:** Pei-Yau Lung, Dongrui Zhong, Xiaodong Pang.

**Funding acquisition:** Jinfeng Zhang.

**Investigation:** Pei-Yau Lung, Jinfeng Zhang.

**Methodology:** Pei-Yau Lung, Dongrui Zhong, Xiaodong Pang, Jinfeng Zhang.

**Project administration:** Jinfeng Zhang.

**Resources:** Jinfeng Zhang.

**Supervision:** Jinfeng Zhang.

**Validation:** Pei-Yau Lung, Dongrui Zhong, Yan Li.

**Visualization:** Pei-Yau Lung, Dongrui Zhong.

**Writing – original draft:** Pei-Yau Lung, Dongrui Zhong.

**Writing – review & editing:** Dongrui Zhong, Xiaodong Pang, Yan Li, Jinfeng Zhang.

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
