## [Decision Letter · Decision Letter 0]

30 Oct 2019

Dear Dr Zhang,

Thank you very much for submitting your manuscript 'Maximizing the Reusability of Gene Expression Data by Predicting Missing Metadata' for review by PLOS Computational Biology. Your manuscript has been fully evaluated by the PLOS Computational Biology editorial team and in this case also by independent peer reviewers. The reviewers appreciated the attention to an important problem, but raised some substantial concerns about the manuscript as it currently stands. While your manuscript cannot be accepted in its present form, we are willing to consider a revised version in which the issues raised by the reviewers have been adequately addressed. We cannot, of course, promise publication at that time.

Please especially address the criticism of reviewer 2 which presents a major hurdle for publication at this time.

Sincerely,

Ilya Ioshikhes

Associate Editor

PLOS Computational Biology

Thomas Lengauer

Methods Editor

PLOS Computational Biology

[LINK]

Reviewer's Responses to Questions

**Comments to the Authors:**

Reviewer #1: attached

Reviewer #2: Although the paper is described as the prediction of missing data in public databases, in essence it is addressing a specific problem of whether specific sample attributes (variables), such as ethnicity, estrogen receptor (ER) status, progesterone receptor status, and similar can be predicted from gene expression data and to what extent. The link to the public data is that if, for instance, a breast cancer gene expression data contains data from patients some of which are ER positive and some ER negative, but the sample status is not given, one can ask if this is possible to derive from the data itself?

One can ask this question within a particular dataset and across datasets. In the first setting one would take a dataset where this variable (e.g., ER status) is included in the annotation, hide it, train a machine learning model and then use cross-validation to assess the accuracy of the model. In the second setting, one would train the model on one dataset, and then test on a different dataset. The second approach makes sense, the first really does not. One could also think of an approach where several datasets are merged, jointly normalised and then model is trained on the merged dataset, but I cannot imagine why one would do this, rather than the second – effectively meta-analysis approach. Unfortunately, I was not able to understand from the paper, which of these approaches the authors take (possibly this can be deduced by studying the supplementary material, but I did not do this, nor should a reader). I think that if the authors are invited to resubmit the paper, they should clarify this.

In addition, I am worried that instead of using well accepted methods for assessing the accuracy of machine learning algorithms, the authors have designed their own methods. I did not fully understand the rationale of this.

Finally, one has to ask what is the utility of this? The authors spend significant space in the manuscript discussing comparisons of various ML methods but is this really interesting? It would be more interesting to choose a wider range of variables (e.g., sex, age bracket, tumour stage, etc) and compare how well each can be predicted (cross datasets). The problem of course would be that most variables are specific to particular diseases, or particular studies.

The authors claim that their missing value imputation approach help in downstream analysis, particularly in differential gene expression. It is not surprising that it does (though I was not able to easily understand the author proof of this), but one could just as well use a latent variable approach (i.e., assume that there are latent variables in the dataset, which need to be factored out) for this problem, an approach which is well familiar to statistical geneticists. Computationally imputing missing values in the public databases would be rather dangerous, as this might easily lead to circularity in the downstream analysis by other authors.

To summarise, I do not think that the paper is written clearly enough, I think the manuscript contains too much philosophy around, e.g., FAIR principles, which are not really relevant to the main line of the presented research, and in the current form it does not represent a clear advancement of science.

Reviewer #3: In this study, we develop a framework to predict missing metadata of gene expression datasets to maximize their reusability. We propose a new metric called Proportion of Cases Accurately Predicted (PCAP), which is optimized in our specifically-designed machine learning pipeline.

The problem is extremely important and relevant of ensuring high quality metadata and with the increasing amount of data, methods are required to tackle it. The paper proposes a method to predict missing metadata using machine learning methods and show that their predictions can be used in downstream analysis. However, I have concerns about the main research question, methodology design and evaluation.

Here are my detailed comments:

- “However, the majority of public genomic data do not contain enough metadata, which severely limits their reusability.” Please provide evidence for this.

- “It may seem quite straightforward to infer missing metadata using gene expression profiles, given the abundance of previous research works.” Claims like these must be supported by evidence.

- “In this study, we investigate the above issues to infer missing metadata using multiple gene expression datasets with a wide variety of machine learning methods.” This is too broad and vague. Which machine learning methods? Why those methods? Why those specific datasets?

- Overall, the main research question/hypothesis is missing.

- Another major concern is that there is no discussion of related work. What is the gap that the authors are trying to fill? There have been studies that have looked at methods to predict metadata:

-- Predicting biomedical metadata in CEDAR: A study of Gene Expression Omnibus (GEO) https://www.sciencedirect.com/science/article/pii/S1532046417301405

-- Predicting structured metadata from unstructured metadata https://academic.oup.com/database/article/doi/10.1093/database/baw080/2630448

-- massiR : a method for predicting the sex of samples in gene expression microarray datasets https://academic.oup.com/bioinformatics/article/30/14/2084/2390865

- This sentence “For each variable, the subset of samples with missing values was not considered in the analysis of the corresponding variable, since we cannot evaluate the performance of our method if we use these samples.” puzzled me because I thought that the aim of the project was to to infer missing metadata.

- There doesn’t seem to be any novelty of the method that is used apart from an evaluation measure that they propose (PCAP - Proportion of Cases Accurately Predicted), which is also not evaluated. Only PCAP95 and PCAP90 seemed to have evaluated. How were these values determined? What is the threshold at which the accuracy decreases?

- In the “Normalization of gene expression” section, three normalization techniques are mentioned, whereas in the results sections only the effect of Rank Normalization is discussed. The other two methods’ results should also be discussed.

- I miss details on the “Model selection pipeline”, how does that work? There is no explanation of the pipeline.

- The goal of predicting missing metadata is to use the corresponding gene expression data in other analyses.

- The example presented about the “differential gene expression analysis (DGEA) using predicted race” is a bit unclear. I would suggest to create a graphic to help understand it.

- Additionally, it is insufficient to show that the predicted metadata can be useful in only one use case

- “recover a high percentage of samples with the desired accuracy” - please add more details for this

- Also, it is unclear whether the method can be generalized to other metadata

- Additionally, the authors do not provide their code which makes it even more difficult to understand the implementation of the methodology.

**Have all data underlying the figures and results presented in the manuscript been provided?**

Reviewer #1: Yes

Reviewer #2: Yes

Reviewer #3: No: The cleaned metadata is not made available and nor is the code.

PLOS authors have the option to publish the peer review history of their article (what does this mean?). If published, this will include your full peer review and any attached files.

Reviewer #1: Yes: Shannon E. Ellis

Reviewer #2: No

Reviewer #3: Yes: Amrapali Zaveri

---

## [Decision Letter · Decision Letter 1]

30 Jul 2020

Dear Dr. Zhang,

Thank you very much for submitting your manuscript "Maximizing the Reusability of Gene Expression Data by Predicting Missing Metadata" for consideration at PLOS Computational Biology.

As with all papers reviewed by the journal, your manuscript was reviewed by members of the editorial board and by several independent reviewers. In light of the reviews (below this email), we would like to invite the resubmission of a significantly-revised version that takes into account the reviewers' comments.

We cannot make any decision about publication until we have seen the revised manuscript and your response to the reviewers' comments. Your revised manuscript is also likely to be sent to reviewers for further evaluation.

Sincerely,

Ilya Ioshikhes

Associate Editor

PLOS Computational Biology

Thomas Lengauer

Methods Editor

PLOS Computational Biology

Reviewer's Responses to Questions

**Comments to the Authors:**

Reviewer #1: Generally, I would love exploration here and clarity in explanation/display of testing and am unsure how I would *use* this in the future if I found the results compelling. The two main conclusions seem to be:

1. Larger sample size not always the answer after metadata predictions used to supplement these points

2. PCAP a better metric

From the text and figures, I'm not convinced of these. I'm not saying they're untrue...I just struggle to draw these conclusions from the information provided. If these are not to be the takeaways, I'd revisit the text. If these are the main takeaways, I'd revisit the clarity of explanations provided and figures.

Major:

- The manuscript has not become clearer in this round of revision. I'm not convinced that PCAP is a better approach and I'm even less clear about what predictions are being made, what cutoffs to use, how to determine ideal sample size after prediction, or what metrics to use to decide this. I think taking a hard look at the organization of these explanations and clarity in display of results would go a long way.

- Figure 4 is very unclear to me. PCAP90 (the metric used in the manuscript) looks pretty darn close to random to me (on median); What about other metrics that you claim PCAP is better than? Why not included here?

- Figure 6 - without knowing what each point is, it's hard to know what variables you can predict with this method and if a different method would be better

- Thanks for sharing the URL to the code in response to reviewers. Please include this in the manuscript directly.

Minor:

- First paragraph of introduction: "The availability of..." is missing a citation

- Bottom of Page 3 (newly added material) - "It is unknown that whether" <- I think that is an additional word. If not, I'm not sure what this sentence is trying to say. Also, that sentence needs a citation or better wording. Also DGES is the initialism used here, but I think it's supposed to be DGEA

- Page 7 - new content; "Our pipeline can be easily extended to multi-class predictions" <- either demonstrate this directly in manuscript or remove this statement

- Top of pg 10 (new content) - unsure what the sentence "For variable," is conveying - have variables A,B,C, and D been introduced in text outside of figures yet?

- Figure 1 colors - hard to compare between plots as negative and positive and the same color in (a) and (b)

Reviewer #2: The authors have significantly extended the study and have addressed my concerns.

Reviewer #4: Attached file

**Have all data underlying the figures and results presented in the manuscript been provided?**

Reviewer #1: Yes

Reviewer #2: Yes

Reviewer #4: Yes

PLOS authors have the option to publish the peer review history of their article (what does this mean?). If published, this will include your full peer review and any attached files.

Reviewer #1: No

Reviewer #2: No

Reviewer #4: No
---

## [Decision Letter · Decision Letter 2]

9 Oct 2020

Dear Dr. Zhang,

We are pleased to inform you that your manuscript 'Maximizing the Reusability of Gene Expression Data by Predicting Missing Metadata' has been provisionally accepted for publication in PLOS Computational Biology.

Best regards,

Ilya Ioshikhes

Associate Editor

PLOS Computational Biology

Thomas Lengauer

Methods Editor

PLOS Computational Biology

Reviewer's Responses to Questions

**Comments to the Authors:**

Reviewer #1: Thank you to the authors for their thoughtful revisions and updates to the manuscript. I'm happy with the manuscript in its current form and have no further suggestions or requests.

Reviewer #4: No more comment.

**Have all data underlying the figures and results presented in the manuscript been provided?**

Reviewer #1: Yes

Reviewer #4: Yes

PLOS authors have the option to publish the peer review history of their article (what does this mean?). If published, this will include your full peer review and any attached files.

Reviewer #1: No

Reviewer #4: **Yes: **Hulin Wu

---

## [Editor Report · Acceptance letter]

23 Oct 2020

PCOMPBIOL-D-19-01671R2 

Maximizing the Reusability of Gene Expression Data by Predicting Missing Metadata

Dear Dr Zhang,

I am pleased to inform you that your manuscript has been formally accepted for publication in PLOS Computational Biology. Your manuscript is now with our production department and you will be notified of the publication date in due course.

With kind regards,

Matt Lyles
